# Novel Bispecific Aptamer Enhances Immune Cytotoxicity Against MUC1-Positive Tumor Cells by MUC1-CD16 Dual Targeting

**DOI:** 10.3390/molecules24030478

**Published:** 2019-01-29

**Authors:** Zhaoyi Li, Yan Hu, Yacong An, Jinhong Duan, Xundou Li, Xian-Da Yang

**Affiliations:** Institute of Basic Medical Sciences, Chinese Academy of Medical Sciences & Peking Union Medical College, Beijing 100005, China; lizhaoyi@ibms.pumc.edu.cn (Z.L.); hyjy2008@hotmail.com (Y.H.); anyacong@ibms.pumc.edu.cn (Y.A.); jinhong_duan@aliyun.com (J.D.); lixd1012@163.com (X.L.)

**Keywords:** aptamer, bispecific, MUC1, CD16, immunotherapy

## Abstract

A promising strategy in cancer immunotherapy is the employment of a bispecific agent that can bind with both tumor markers and immunocytes for recruitment of lymphocytes to tumor sites and enhancement of anticancer immune reactions. Mucin1 (MUC1) is a tumor marker overexpressed in almost all adenocarcinomas, making it a potentially important therapeutic target. CD16 is expressed in several types of immunocytes, including NK cells, γδ-T cells, monocytes, and macrophages. In this study, we constructed the first bispecific aptamer (BBiApt) targeting both MUC1 and CD16. This aptamer consisted of two MUC1 aptamers and two CD16 aptamers linked together by three 60 nt DNA spacers. Compared with monovalent MUC1 or CD16 aptamers, BBiApt showed more potent avidity to both MUC1-positive tumor cells and CD16-positive immunocytes. Competition experiments indicated that BBiApt and monovalent aptamers bound to the same sites on the target cells. Moreover, BBiApt recruited more CD16-positive immunocytes around MUC1-positive tumor cells and enhanced the immune cytotoxicity against the tumor cells in vitro. The results suggest that, apart from bispecific antibodies, bispecific aptamers may also potentially serve as a novel strategy for targeted enhancement of antitumor immune reactions against MUC1-expressing malignancies.

## 1. Introduction

Globally, cancer is one of the leading causes of death in the 21st century. With the growth of the aging population worldwide, the burden of cancer continues to increase due to growth of incidence [1,2]. Traditional cancer therapies include surgery, chemotherapy, and radiotherapy. In addition, cancer immunotherapy became mainstream treatment in the past decade [3]. The major progresses in cancer immunotherapy are highlighted by chimeric antigen receptor T (CAR-T) cell therapy and immune checkpoint blockade (ICB). CAR-T therapy achieved remarkable success in hematologic malignancy. CD19-directed CAR-T therapy was efficacious in treating relapsed and refractory acute lymphoblastic leukemia (ALL) with very high response rates [4]. However, CAR-T therapy requires extraction of each patient’s autologous T-cells followed by lentivirus treatment to generate tumor-antigen specific T-cells. As a result, CAR-T therapy is extremely expensive due to the costs of T-cells culturing, processing, storage, and transportation. ICB, such as PD-1/PD-L1 antibodies, can stimulate the patient’s immune system and unleash antitumor immune activities. PD-1 antibodies show reproducible benefit in approximately 20% of patients with various previously incurable cancers [5]. Nevertheless, PD-1/PD-L1 antibodies do not represent a targeted antitumor immunotherapy, and the raised immune activities are not tumor-specific. Therefore, autoimmune side effects were observed in many clinical trials, and some of them even had lethal outcomes [6]. Currently, there are many studies aimed at improving CAR-T or ICB therapies. Simultaneously, novel types of immunotherapies are being explored and developed.

In addition to CAR-T and ICB therapies, bispecific antibody (BiAb) is also a promising strategy of cancer immunotherapy. BiAb is made of a tumor-aiming antibody linked to an immunocyte-aiming antibody, and can bind with both the tumor cells and the lymphocytes. Thus, a specific population of lymphocytes can be recruited to the tumor cells and exhibit a powerful antitumor immune reaction. Both BiAb and CAR-T represent tumor-targeted therapies and recruit lymphocytes to tumor cells, while BiAb has a major advantage as a product that can be readily used to treat many patients without the intricate cell culturing and genetic engineering process for every individual patient required in CAR-T therapy. Moreover, while both BiAb and ICB are implemented by antibodies, BiAb is a targeted therapy and does not boost immune reaction nonspecifically as in ICB therapy, therefore drastically reducing autoimmune side effects. The first FDA-approved BiAb, Blinatumomab, targets both CD3 and CD19 and achieved potent antitumor immune responses [7]. Clinical trial NCT02013167 compared blinatumomab and chemotherapy for advanced acute lymphoblastic leukemia and found that the remission rates within 12 weeks were 34% in blinatumomab group and 12% in chemotherapy group [8]. The above indicates that bispecific agent represents a strategy of antitumor immunotherapy with great promise. However, CD19 is mainly overexpressed in hematologic malignancies, including most B-cell lymphomas, B-cell acute lymphocytic leukemia, and B-cell chronic lymphocytic leukemia [7]. In order to further advance the benefit of bispecific cancer immunotherapy, we need to aim at a tumor marker that is more widely expressed than CD19.

MUC1 is a tumor marker that may serve as a potential target for treatment of a wide variety of cancers. MUC1 is a transmembrane protein with a heavily glycosylated extracellular domain. It is normally expressed in glandular and luminal epithelial cells of many tissues, protecting the underlying epithelia from pollutants or microbes. In cancer cells, MUC1 is either hypo-glycosylated or aberrantly glycosylated, resulting in the exposure of tumor-specific peptide or carbohydrate epitopes [9]. The aberrant MUC1 is overexpressed in almost all human adenocarcinomas, including lung, colon, pancreatic, breast, ovarian, and prostate cancers [10]. Studies show that MUC1 is associated with tumor initiation and progression [11]. Highly expressed MUC1 is also linked with poor prognosis [9]. The differences of MUC1 between normal and cancer cells make it a promising target for cancer therapy. Several clinical trials are ongoing to probe the potential of MUC1-based products [12]. Because MUC1 is overexpressed in many malignancies, targeting MUC1 appears to be an appropriate strategy for designing bispecific agent with broader range of applications.

Apart from antibody, aptamers can also be used to construct bispecific agents. Aptamers are short, single-stranded DNA or RNA (ssDNA or ssRNA) that can form complicated three-dimensional structures and bind to target molecules with high affinity and specificity. Aptamers are usually selected from a very large pool of ssDNA or ssRNA using a method called systematic evolution of ligands by exponential enrichment (SELEX). Compared with antibodies, aptamers have smaller size and can penetrate tumor tissues to bind with some hidden domains that are inaccessible for antibodies [13]. Most aptamers can be chemically synthesized with low production cost. Moreover, aptamers have relatively low immunogenicity [14]. Macugen is the first FDA-approved aptamer-based therapeutics for treating wet age-related macular degeneration, demonstrating the clinical application potential of aptamers [15]. The selected aptamers through SELEX procedure are commonly monovalent and may not always have satisfactory target-binding capability. To enhance the binding avidity, multivalency is a feasible strategy. A multivalent aptamer is defined as a construct composed of two or more aptamers of the same kind and usually has significantly higher target-binding avidity compared with the monovalent aptamer [16].

In this study, we constructed the first bispecific aptamer targeting both MUC1 and CD16. CD16 is expressed in many types of immunocytes, including NK cells, γδ-T cells, monocytes, and macrophages. The goal of this bispecific aptamer is to engage CD16-positive immunocytes with MUC1-positive cancer cells in order to selectively enhance the antitumor cytotoxicity. To further improve the binding avidity, two MUC1 aptamers and two CD16 aptamers were integrated into one construct to create a bivalent bispecific aptamer (BBiApt). We now report that BBiApt recruited lymphocytes to A549 cancer cells and facilitated the antitumor immune cytotoxicity in vitro.

## 2. Results

### 2.1. Design of BBiApt

In order to engage immunocytes with tumor cells, we designed a bispecific aptamer that can bind with both MUC1 and CD16. The MUC1 aptamer was selected by our lab previously [17], and the CD16 aptamer was selected by Achim Boltz et al. [18]. A bispecific aptamer made of MUC1 and CD16 aptamer should, in principle, tether together CD16-positive immunocytes and MUC1-positive tumor cells. In order to strengthen the binding and recruit more cells, multivalent binding is preferred. Hence, two MUC1 aptamers and two CD16 aptamers were adopted to generate bivalent bindings in this study (Figure 1A). The secondary structure of the BBiApt was predicted using UNAFold tools (Figure 1B). To avoid unwanted interactions between two aptamers if they closely linked, oligonucleotide containing 60 bases was used as a spacer to conjugate these two aptamers into one single sequence. The entire length of this structure was 420 nucleotides (nt), including two MUC1 aptamers, two CD16 aptamers, and three 60 nt ssDNA spacers.

### 2.2. Production of BBiApt

The designed BBiApt had a total of 420 nucleotides. Single-strand DNA of this length was difficult to synthesize chemically and had to be produced using other approaches. Cosimo Ducani et al. developed an enzymatic method to produce elongated ssDNA oligonucleotides [19]. In this study, we adopted the method to produce BBiApt. Briefly, the BBiApt sequence was cloned into PUC118 plasmid, which was used to transfect JM109 E. coli. The single-strand phagemid DNA containing BBiApt was amplified through E. coli and helper phage co-culturing, extracted, and digested by restriction endonuclease to generate BBiApt. To evaluate whether the phage production system generated the desired DNA and whether the endonuclease cut out the correct product of BBiApt, DNA extracted from helper phage and its digested counterpart were analyzed by electrophoresis in denaturing polyacrylamide gel. Compared to the undigested DNA, the digested DNA had an obvious low MW band that represented the BBiApt. Notably, there were larger DNA segments that presumably represented the remaining phagemid DNA after the endonuclease digestion (Figure 2). The BBiApt band was extracted from PAGE gel and used for subsequent experiments.

### 2.3. BBiApt‘s Binding to Target Cells

Although BBiApt was based on MUC1 and CD16 aptamers, whether this new DNA structure retained the capability to bind with MUC1 and CD16 was unknown. It was thus important to reevaluate BBiApt’s binding to the target cells. Moreover, it was also necessary to investigate whether the bivalent BBiApt had higher binding avidity to target cells compared to monovalent MUC1 or CD16 aptamers. To address these issues, MUC1-positive A549 cells and MUC1-negative HepG2 cells were incubated with BBiApt or monovalent MUC1 aptamer and evaluated by flow cytometry. As shown in Figure 3A,C, both BBiApt and MUC1 aptamer bound with MUC1-positive A549 cells, but not with MUC1-negative HepG2 cells. Moreover, BBiApt had a stronger binding to A549 cells than monovalent MUC1 aptamer. A similar study was also conducted to compare the bindings of BBiApt vs. monovalent CD16 aptamer to CD16-positive cells (PBMC) and CD16-negative cells (Jurkat). Numerous studies show that PBMC are rich in CD16-positive cells, including NK cells, monocytes, and macrophages [20,21,22]. Therefore, PBMC are routinely used as CD16-positive immunocytes in antitumor immunological studies in vitro [18]. As shown in Figure 3B,D, both BBiApt and CD16 aptamer bound with CD16-positive PBMC, but not with CD16-negative Jurkat cells. BBiApt also had a stronger binding to PBMC than monovalent CD16 aptamer. Taken together, the above results indicated that higher target avidities were generated by the bivalent aptamer compared with the monovalent aptamers.

### 2.4. BBiApt Bound to the Same Targets as the Monovalent Aptamers

Although BBiApt showed good avidity for both target cells, whether this new ssDNA structure bound to the same target domains as the monovalent aptamers was uncertain. To study this issue, BBiApt was mixed with monovalent aptamers in a competition experiment. Specifically, MUC1-positive A549 cells were incubated with FAM-labeled MUC1 aptamers with or without BBiApt. Similarly, CD16-positive PBMCs were incubated with FAM-labeled CD16 aptamers with or without BBiApt. As shown in Figure 4, the binding of monovalent aptamers to their target cells significantly decreased when BBiApt were presented in both experiments. The results indicated that BBiApt and the monovalent aptamers probably bound to the same targets.

### 2.5. The Target Molecules of BBiApt were Membrane Proteins

In order to bring tumor cells and immunocytes together, it was necessary for BBiApt to bind with the membrane proteins on the surface of both types of cells. To evaluate whether BBiApt indeed bound the extracellular domains of membrane proteins, trypsin digestion experiments were performed [23,24]. Specifically, target cells were treated with trypsin, incubated with BBiApt, and evaluated by flow cytometry. As shown in Figure 5, trypsin treatment resulted in a significant setback of fluorescent signals, indicating that the binding site of BBiApt was probably the extracellular domain of membrane proteins.

### 2.6. BBiApt Recruited CD16-positive Immunocytes Around A549 Tumor Cells

The above observations revealed that BBiApt showed good avidity to both A549 cells and PBMCs. However, it was still unknown whether BBiApt was able to tether the two types of cells together. To explore this, live A549 or HepG2 cells were stained green by CFSE, and mixed with live PBMCs that were stained red by eFluro670. The cell mixtures were incubated for 30 min in the presence or absence of BBiApt, and washed thrice to remove the unattached PBMCs. The remaining cell mixtures were observed under fluorescent microscope. As shown in Figure 6, in the presence of BBiApt, more immunocytes gathered around MUC1-positive A549 cells, while few immunocytes were observed around MUC1-negative HepG2 cells. These results indicated that BBiApt could recruit more CD16-positive immunocytes to the vicinity of MUC1-positive tumor cells.

### 2.7. BBiApt Enhanced Cytotoxicity to MUC1-Positive Tumor Cells by CD16-Positive Immunocytes

Although BBiApt could recruit more CD16-positive cells around MUC1-positive tumor cells, it was still unknown whether these recruited immunocytes could enhance the cytotoxicity against the tumor cells. To study this issue, A549 cells or HepG2 cells were co-cultured with PBMCs and treated with MUC1 aptamers, CD16 aptamers, or BBiApt. The immune cytotoxicity in each group was evaluated by measuring the viability of tumor cells with a standard MTS assay. The results were presented in Figure 7. In MUC1-positive A549 cells (Figure 7A), BBiApt significantly enhanced CD16-positive cells’ cytotoxicity against tumor cells, while MUC1 aptamers or CD16 aptamers alone failed to generate a similar effect. In MUC1-negative HepG2 cells (Figure 7B), BBiApt was unable to enhance the immune cytotoxicity. It should be noted that BBiApt *per se* showed no toxicity to tumor cells in the absence of PBMCs. These data indicated that BBiApt could enhance the cytotoxicity of CD16-positive cells toward MUC1-positive tumor cells, but not that toward the MUC1-negative cells.

## 3. Discussion

In this study, a bispecific aptamer was constructed to bind with both MUC1-positive tumor cells and CD16-positive lymphocytes in order to bring the two types of cells together for enhancement of antitumor reaction. We integrated two MUC1 aptamers and two CD16 aptamers into a single construct (BBiApt) and used helper phages to produce enough amount of this relatively large single-strand DNA for subsequent studies (Figure 1 and Figure 2). The bivalent BBiApt showed higher avidity to target cells compared with monovalent MUC1 or CD16 aptamers (Figure 3 and Figure 4). BBiApt was found to bind with the extracellular domains of membrane proteins of MUC1- or CD16-positive cells (Figure 5), and could recruit more CD16-positive immunocytes around the MUC1-positive tumor cells (Figure 6). Moreover, BBiApt selectively enhanced the immune cytotoxicity against the MUC1-positive tumor cells, but not that against the MUC1-negative control cells in vitro (Figure 7). These observations suggest that BBiApt may have application potential for selective enhancement of immunocyte-mediated antitumor reaction against MUC1-positive tumor cells.

Cancer immunotherapy attracted great attention in recent years. CAR-T therapy and PD-1/PD-L1 monoclonal antibodies represent the two main progresses in cancer immunotherapy field. Although CAR-T therapy demonstrated excellent efficacy against CD19-positive malignancies [4,25], its wide application in clinical practice is limited by certain factors, including individualized culturing of genetically engineered lymphocytes for every single patient, high production cost, as well as difficulties for quality control and logistics. PD-1/PD-L1 antibodies showed efficacies in approximately 20% of patients diagnosed with advanced tumors in clinical trials. However, they are not targeted tumor therapy and boost up immune reactions nonspecifically. As a result, they have weak efficacies against many tumors and are associated with various autoimmune side effects, including pneumonitis, hepatitis, colitis, thyroiditis, and hypophysitis [26]. In addition to CAR-T therapy and PD1/PD-L1 antibodies, bispecific agent also represents a promising strategy of cancer immunotherapy. Most of the bispecific agents are bispecific antibodies (BiAb), which can bind with both tumor cells and lymphocytes to bring them together, facilitating a targeted and relatively specific antitumor immune reaction. Bispecific agents have certain advantages for cancer immunotherapy. Compared with CAR-T, bispecific agents are mass produced in factory, do not need to genetically modify lymphocytes for every patient, and therefore are more suitable for wider clinical applications. Compared with PD-1/PD-L1 antibodies, bispecific agents can generate a targeted antitumor immune reaction, and thus avoid the autoimmune side effects generated by nonspecific immune-boosting effects of PD-1/PD-L1 antibodies. Due to these advantages, several BiAbs are under clinical development. Some of them show good efficacies in clinical trials. The first FDA-approved BiAb is Blinatumomab, which showed great efficacy in patients with B-ALL. All in all, bispecific agents show unique characteristics and have broad application prospects in the future cancer immunotherapy field.

The application range of bispecific agents depends on its target selection. Here, we chose MUC1 as the target for the bispecific aptamer because MUC1 is a broad-spectrum tumor marker overexpressed in most adenocarcinomas, including 96.7% of invasive lung cancers; 90% of prostate, pancreatic, and epithelial ovarian tumors; 70% of breast cancers; and even 60% of captured circulating tumor cells from a variety of metastatic cancers [27]. As MUC1 is mainly overexpressed in solid tumors, it is important for bispecific agent to penetrate into tumor tissues in order to serve its function. Previous studies showed that the tumor-targeting capability of monoclonal antibody is often limited due to its insufficient penetration into tumor tissues [28]. Aptamer was reported to have better tumor tissues penetration compared with antibody [13]. Moreover, aptamers are relatively easy to synthesize and modify and exhibit low immunogenicity in vivo. Until now, bispecific aptamer targeting MUC1 was not reported. Here, we constructed a bispecific aptamer that could recruit CD16-positive immunocytes to MUC1-positive tumor cells and selectively enhanced the antitumor cytotoxicity. The results indicate that, in addition to antibodies, aptamers may also serve as the material for building bispecific agent against MUC1-positive tumors.

Multimerization is a strategy to enhance monoclonal antibodies’ tissue specificity and even provide novel functionality [29]. However, the production of multivalent monoclonal antibodies is technically difficult. In contrast, multivalent aptamers are relatively convenient to fabricate. Nevertheless, until now, most of the bispecific aptamers reported in the literature were monovalent. Achim Boltz et.al designed a monovalent bispecific aptamer targeting CD16-cMet, which enhanced cellular antitumor cytotoxicity [18]. Eli Gilboa et al. engineered bispecific aptamers recognizing various targets including PSMA-4-1BB [30] and MRP1-CD28 [31] and found that these agents potentiated immune response and inhibited tumor growth. So far, multivalent bispecific aptamers engineered to enhance the avidities to both tumor and immunocyte have not been reported. In this study, we constructed the first bivalent bispecific aptamer BBiApt, which was made of two MUC1 aptamers and two CD16 aptamers, so that the bindings to both tumor and lymphocyte were strengthened. This construct was able to recruit more CD16-positive cells around MUC1-positive tumor cells and enhanced the antitumor cytotoxicity. The possible mechanism involved was that as immunocytes were drawn to tumor cells by BBiApt, the chance of interaction between the two types of cells greatly increased, activating the lymphocytes and enhancing the antitumor reaction. These results indicate that multivalent aptamers may potentially be used to construct novel bispecific ligands. Specifically, multivalent bispecific aptamers targeting MUC1 and lymphocytes may serve as a new strategy for treatment of most adenocarcinomas. Admittedly, this work is an early-stage attempt to explore the use of multivalent bispecific aptamer for inducing targeted tumor cell lysis. Further work may focus on its application in vivo with animal studies. Additionally, chemical modification for improving the structure’s nuclease-resistant capability is also warranted. 

In summary, we designed a bivalent bispecific aptamer by integrating two MUC1 aptamers and two CD16 aptamers into a single construct. This construct had higher avidities to both MUC1-positive tumor cells and CD16-positive immunocytes, tethered the two types of cells together, and significantly enhanced the antitumor cytotoxicity in vitro. The results indicate that aptamer-based multivalent bispecific agents may potentially serve as a new approach to selectively boost antitumor immune reactions.

## 4. Materials and Methods

### 4.1. Generation of BBiApt

Monovalent MUC1-CD16 bi-specific aptamer was fabricated using overlap-PCR with the following 4 oligonucleotides (synthesized by Invitrogen, Shanghai, China): 

5′-AACCGCCCAAATCCCTAAGAGTCGGACTGCAACCTATGCTATCGTTGATGTCTGTCCAAGCAACACAGACACACTACACACGCACA-3′ (MUC1 aptamer); 

5′-CGTATAGACCCCCGCAGTGGTTGGTGTTTGTGTTTGGTGGTGTTTGGTTTTGTGTTGTGTTGTTTGGTTTGTTGTTGTTTTGTGCGTGTGTAGTGTGTCT-3′; 

5′-CCACTGCGGGGGTCTATACGTGAGGAAGAAGTGG-3′ (CD16 aptamer); 

5′-TTGGTGTTTGTGTTTGGTGGTGTTTGGTTTTGTGTTGTGTTGTTTGGTTTGTTGTTGTTTCCACTTCTTCCTCACGTATA-3′

The underlined parts indicated the linkers between the aptamers. After overlap-PCR, the DNA was cloned into pEASY-T1 vector (Transgen Biotech, China), sequenced, and analyzed to identify the correctly constructed monovalent bi-specific aptamer.

Bivalent MUC1-CD16 bi-specific aptamer was fabricated by linking two monovalent bispecific aptamers with overlap-PCR. Specifically, the monovalent bi-specific aptamer was PCR amplified using the forward primer 5′-AACCGCCCAAATCCCTAAGAGTC-3′ and the reverse primer 5′-biotin-TTGGTGTTTGTGTTTGGTGGTGTTTGGTTTTGTGTTGTGTTGTTTGGTTTGTTGTTGTTTCCACTTCTTCCTCACGTATA-3′. Single-strand monovalent aptamer was obtained using the streptavidin-coated magnetic beads (Promega, Madison, WI, USA) per manufacturer’s instruction. For overlap-PCR, single-strand monovalent aptamer was mixed with the oligonucleotide 5′- TCTTAGGGATTTGGGCGGTTTTGGTGTTTGTGTTTGGTGGTGTTTGGTTTTGTGTTGTGTTGTTTGGTTTGTTGTTGTTTCCACTTCTTCCTCACGTATA-3′ in the presence of high fidelity PCR mix (2x Phanta Master Mix, Vazyme, Nanjing, China). After 10~17 amplifying cycles, the DNA was cloned into pEASY-T1 vector, sequenced, and analyzed to identify the correctly constructed bivalent bi-specific aptamer. To add two hairpins containing EcoRI restriction enzyme digestion sites, PCR was performed with bivalent bi-specific aptamer as template, using the forward primer 5′-GAATTCTGTCAAAAAGACAGAATTCAACCGCCCAAATCCCTAAGA-3′ and the reverse primer 5′-GAATTCCAAGTTTTTCTTGGAATTCCCACTTCTTCCTCACGTATA-3′ (EcoRI digestion sites were underlined). The DNA was cloned into phagemid (PUC118) vector (ClonExpress Ultra One step Cloning Kit, Vazyme, Nanjing, China), sequenced, and analyzed to identify the correctly constructed bivalent bi-specific aptamer with two EcoRI digestion sites.

Single-strand BBiApt was produced using M13KO7 helper phage. *E. coli* JM109 competent cells (TaKaRa, Kusatsu, Japan) were transfected with phagemid (PUC118) containing the BBiApt sequence smeared onto a culture dish and cultured overnight. A single colony was picked and inoculated in 50 mL TB medium (containing penicillin 100 U/mL), grown at 37 °C and 250 rpm until slightly turbid. The bacteria were mixed with 50 μL M13KO7 helper phages (NEB, Ipswich, MA, USA) and cultured under shaking for 90 min. The mixture was treated with kanamycin (final concentration of 75 μg/mL) and grown overnight at 37 °C and 250rpm. The bacteria were removed by centrifuging twice at 14,000 rpm for 10 min. The supernatant was mixed with 0.2 volume of 2.5 M NaCl/20% PEG-8000 and incubated at 4 °C for 60 min. The mixture was centrifuged at 14,000 rpm for 15 min at 4 °C to get the phage pellet. The pellet was resuspended in 2 mL Tris (10 mM, pH 8.5) and centrifuged at 5000 rpm for 5 min to remove any remaining bacteria. The supernatant was mixed with 4 mL 0.2 M NaOH/1% SDS, gently swirled for 3 min, and treated with 3 mL potassium acetate (3 M, PH 5.5). The mixture was incubated at 4 °C for 10 min and centrifuged at 14,000 rpm for 30min at 4 °C. Supernatant was mixed with 2–3 volumes of 100% ethanol, incubated at 4 °C for 1 h, and centrifuged at 14,000 rpm for 30 min at 4 °C to obtain the single-strand DNA pellet. The DNA pellet was washed twice using 70% ethanol and resuspended in 1mL Tris (10 mM, pH 8.5). The concentration of ssDNA was measured by nanodrop 2000 (ThermoFisher, Waltham, MA, USA). Single-strand DNA (50 ng/μL) products were digested with EcoRI (0.5 U/μL, ThermoFisher, Waltham, MA, USA) in 2× Tango buffer by incubating at 37 °C for 3 h followed by heat-inactivation at 80 °C for 20 min. The digestion products were analyzed by PAGE (6% polyacrylamide, 8 M urea mixed in 1× TBE buffer) at 120 V for 45 min. Gel was stained with SYBR GOLD (1×; Invitrogen) for 15 min and the BBiApt product was purified from the PAGE gel.

### 4.2. Cell Cultures

A549 (human lung cancer), HepG2 (human liver cancer), and Jurkat (human T-cell lymphoma) cells were obtained from the Cell Center of Chinese Academy of Medical Science (Beijing, China). A549 and HepG2 cells were cultured in DMEM medium. Jurkat cells were cultured in RPMI 1640 medium. Both media were supplemented with 10% fetal bovine serum (FBS), 100 U/mL penicillin, and 100 μg/mL streptomycin. Peripheral blood mononuclear cell (PBMC) were isolated from healthy donors by using lymphocyte separation medium (TBD, Tianjin, China) following the manufacturer’s protocol. All donors were required to sign an informed consent form according to procedures approved by the Ethics Committee at Chinese Academy of Medical Sciences and Peking Union Medical College. All cells were grown at 37 °C with 5% CO_2_.

### 4.3. Cellular Binding Assays

A549 and HepG2 cells (2 × 10^5^) were gently scraped and washed with PBS twice. The cells were suspended in 250 μL of PBS, then incubated for 30 min with gentle shaking with equimolar (40 pmol) FAM-labeled probe, FAM-labeled MUC1 aptamer, or FAM-labeled BBiApt, which was generated by incubating BBiApt with FAM-labeled probe (5′FAM-TGTTTGGTTTTGTGTTGTGT-3′) is complementary to the spacer sequence in BBiApt. PBMC and Jurkat cells (2 × 10^5^) were washed with PBS twice, suspended in 250 μL of PBS, and incubated with equimolar FAM-labeled probe, FAM-labeled CD16 aptamer, or FAM-labeled BBiApt for 30 min with gentle shaking. All cells were washed thrice with 250 μL PBS and analyzed by flow cytometry (Accuri C6, BD, San Jose, CA, USA). Experiments for binding assays were repeated three times.

### 4.4. Binding Competition Studies

A549 cells and PBMCs (2 × 10^5^) were incubated with 40 pmol FAM-labeled MUC1 aptamer or CD16 aptamer at 37 °C for 30 min, and then treated with 40 pmol BBiApt for another 30 min, to compete with the FAM-labeled aptamers. All cells were washed thrice with 250 μL PBS and analyzed by flow cytometry. Experiments for binding competition studies were repeated three times.

### 4.5. Trypsin Digestion Experiment

A549 cells and PBMCs (2 × 10^5^) were washed with PBS twice and resuspended in 250 μL of PBS. Two hundred μL 0.05% trypsin/0.02% EDTA was added to the cells at 37 °C. Ten minutes later, pure fetal bovine serum was added to quench the trypsin. All cells were washed thrice, resuspended in 250 μL PBS, and incubated with 40 pmol FAM-labeled BBiApt for 30 min. Then all cells were washed thrice with 250 μL PBS and analyzed by flow cytometry. Experiments for trypsin digestion were repeated three times.

### 4.6. Confocal Imaging Studies

A549 and HepG2 (1 × 10^4^) tumor cells were grown in Lab-Tek Chamber Slide System (ThermoFisher, Waltham, MA, USA). Tumor cells were dyed with CFSE for 10 min, while PBMCs (5 × 10^4^) were dyed with eFluro670 for 10 min (ThermoFisher, Waltham, MA, USA). All cells were washed with culture medium thrice and resuspended in PBS. PBMC suspended in PBS with or without 5 pmol BBiApt were added to the tumor cell chambers, which were incubated at 37 °C for 30 min with gentle shaking. Chambers were washed with PBS for three times to remove the unattached PBMCs. Confocal laser scanning microscopy was applied to evaluate the spatial relations between tumor cells and immunocytes.

### 4.7. In Vitro Cytotoxicity Assays

A549 and HepG2 tumor cells were grown in 96-well plates. After adding PBMC at effector:target ratio (E:T) of 20:1 to the tumor cells, the mixture was treated with 5 pmol of free MUC1 aptamer, free CD16 aptamer, a mixture of free MUC1 and CD16 aptamers, BBiApt, or probe (5′-TGTTTGGTTTTGTGTTGTGT-3′). The cellular mixture was incubated in 37 °C for 36 h. Tumor cells were washed twice with PBS. MTS assay (Promega, Madison, WI, USA) was used to determine the cell viability according to the standard protocol as outline by the manufacture. Experiments for in vitro cytotoxicity assays were repeated three times.

### 4.8. Statistics

Statistical analysis was performed with the statistical SPSS 22.0 software. One-way ANOVA with Fisher’s least significant difference (LSD) post hoc comparisons at 99% confidence interval was used to calculate the significant differences among the groups. All data were presented as a mean value with its standard deviation indicated (mean ± SD).

## Figures and Tables

**Figure 1 molecules-24-00478-f001:**
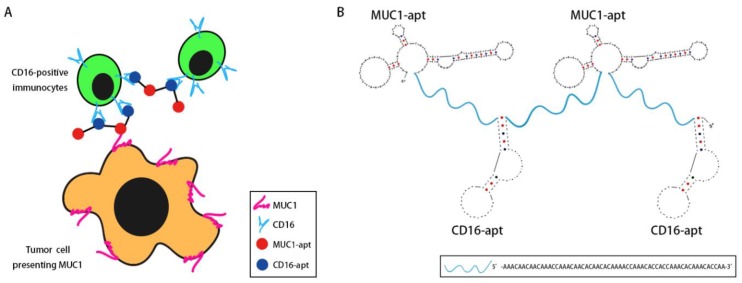
Design of bivalent bispecific aptamer (BBiApt): (**A**) Scheme of BBiApt recruiting CD16-positive immunocytes to MUC1-positive tumor cells. (**B**) The predicted secondary structure of BBiApt using UNAFold tools.

**Figure 2 molecules-24-00478-f002:**
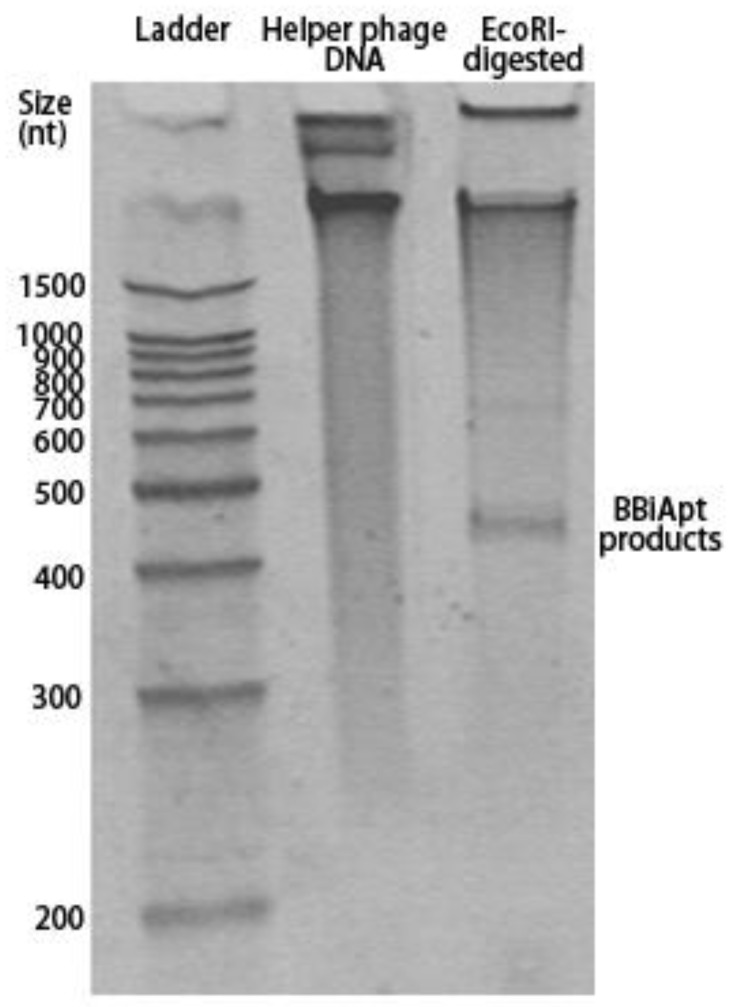
Digestion of the amplified phagemid DNA-released BBiApt products. **Lane 1**: 1500bp ladder; **Lane 2**: Phagemid DNA; **Lane 3**: Digested phagemid DNA. The BBiApt products band could be seen above the 400bp ladder mark. The denaturing polyacrylamide gel was stained by SYBR GOLD.

**Figure 3 molecules-24-00478-f003:**
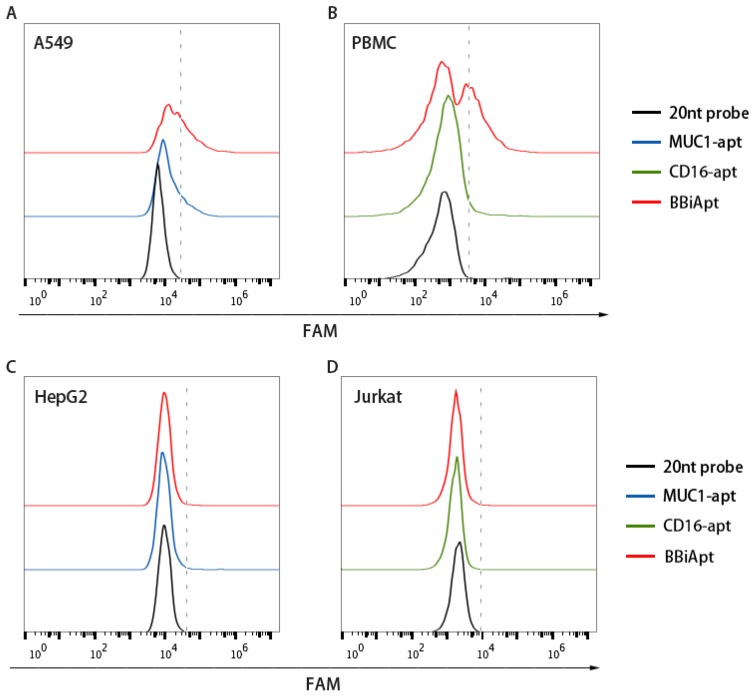
Bindings of BBiApt and monovalent MUC1 or CD16 aptamers to target or control cells. A FAM-labeled 20 nt probe that hybridized to the nonfunctional part of BBiApt was utilized to generate fluorescent signal for BBiApt. This 20 nt probe also served as the background control. MUC1-positive A549 cells (**A**), CD16-positive PBMCs (**B**), MUC1-negative HepG2 cells (**C**), or CD16-negative Jurkat cells (**D**) were treated with the probe (black lines), monovalent aptamers (blue and green lines), or BBiApt (red lines), and evaluated by flow cytometry.

**Figure 4 molecules-24-00478-f004:**
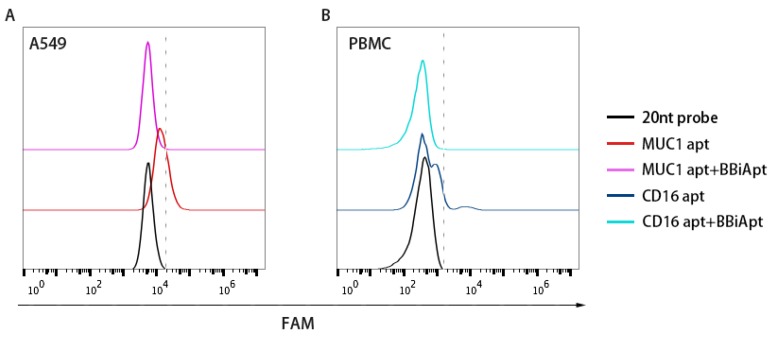
The bindings of monovalent aptamers to MUC1-positive A549 cells (**A**) or CD16-positive PBMCs (**B**) in the presence or absence of BBiApt. Red line represented A549 cells treated with FAM-labeled MUC1 aptamer. Purple line represented A549 cells treated with both FAM-labeled MUC1 aptamer and BBiApt. Dark blue line represented PBMCs treated with FAM-labeled CD16 aptamer. Light blue line represented PBMCs treated with both FAM-labeled CD16 aptamer and BBiApt. Black line represented cells treated with a FAM-labeled 20 nt DNA which served as a background control.

**Figure 5 molecules-24-00478-f005:**
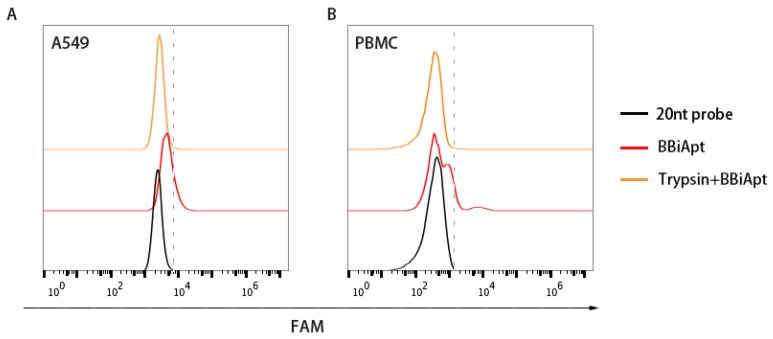
Bindings of BBiApt to A549 cells (**A**) and PBMCs (**B**) after trypsin treatment. A FAM-labeled 20 nt probe that hybridized to the nonfunctional part of BBiApt was utilized to generate fluorescent signal for BBiApt. Red line represented cells treated with BBiApt. Orange line represented cells treated with trypsin for 10 min first, followed by incubation with BBiApt. Black line represented cells treated with the FAM-labeled 20 nt probe which served as a background control.

**Figure 6 molecules-24-00478-f006:**
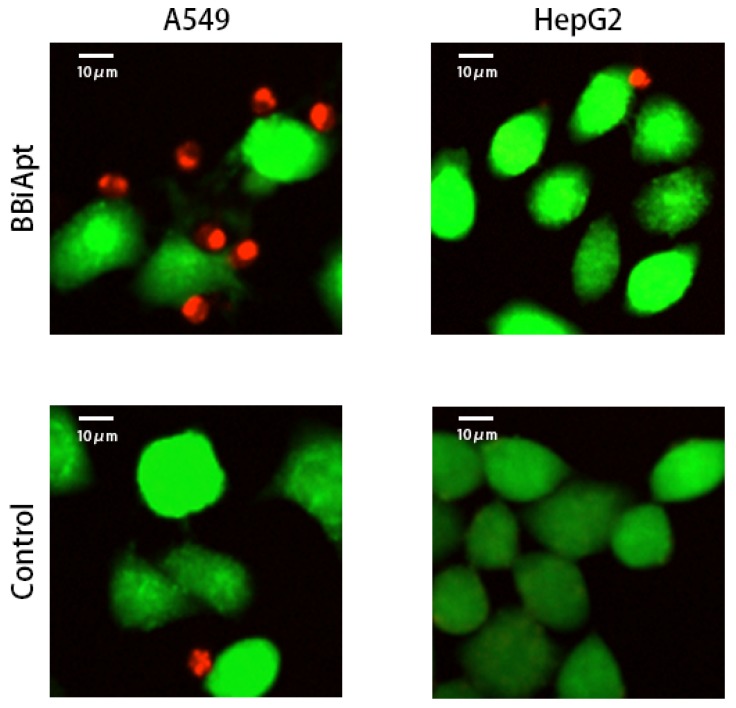
Recruitment of immunocytes to target cells in the presence or absence of BBiApt. Live MUC1-positive A549 cells or MUC1-negative HepG2 cells were stained green with CFSE, while live PBMCs were stained red with eFluro670. Tumor cells and PBMCs were mixed together, with or without BBiApt. After washing thrice, the remaining cells were observed with laser scanning confocal microscope.

**Figure 7 molecules-24-00478-f007:**
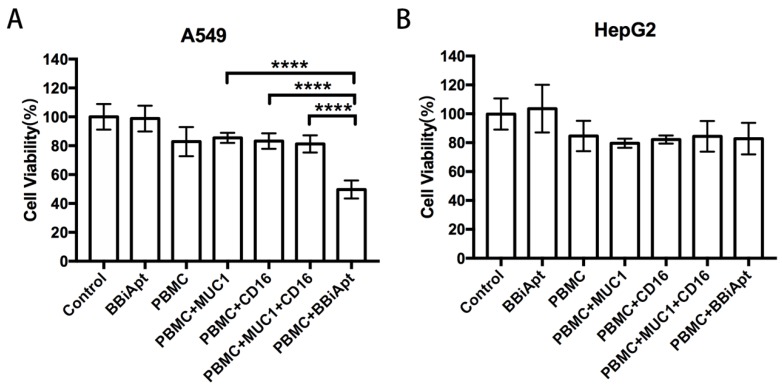
The immune cytotoxicity to MUC1-positive A549 cells or MUC1-negative HepG2 cells. Tumor cells were co-cultured with PBMCs, received various treatments, washed, and evaluated for cell viability with standard MTS assays. (**A**) MUC1-positive A549 cells treated with BBiApt alone, PBMC alone, PBMC plus free MUC1 aptamers, PBMC plus free CD16 aptamers, PBMC plus free MUC1 aptamers and free CD16 aptamers, or PBMC plus BBiApt. (**B**) MUC1-negative HepG2 cells treated in the same way as the A549 cells.

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
