# Peer review of "Novel Bispecific Aptamer Enhances Immune Cytotoxicity Against MUC1-Positive Tumor Cells by MUC1-CD16 Dual Targeting"

_molecules, 2019, doi:10.3390/molecules24030478_

Round 1

Reviewer 1 Report

The authors describe the design of a bi-specific aptamer consisting of 2 different aptamers, one against a tumor marker that appears in many types of tumors (MUC1) and another against a protein present in immunocytes (CD16). The article demonstrates that this complex is able to bind both cell types in a specific way and capable of producing the death of tumor cells by the effect of immunocytes located in their proximity.

The article is very interesting and the work is clearly presented. I only comment a few suggestions:

1. Correct errors in the text:

         Page 3, line 112. Change "et.al" by "et al."

         Page 11, line 339. Enter a space in "treated with"

         Page 12, line 348. Correct "sing-stranded"

         Page 12, line 362. Correct "CO2"

2. I think it would be convenient to put throughout the text the times that each experiment has been done. In my opinion, many of the results presented show slight differences between the different treatments and only the fact that the result is repeated in a sufficient number of experiments gives it credibility.

Author Response

Response to Reviewer 1 Comments

Point 1: Correct errors in the text:

Page 3, line 112. Change "et.al" by "et al."

Page 11, line 339. Enter a space in "treated with"

Page 12, line 348. Correct "sing-stranded"

Page 12, line 362. Correct "CO2"

Response 1: Thank you very much for the valuable comments that help us to improve the manuscript. Accordingly, we have corrected the errors in the revised manuscript.

Point 2: I think it would be convenient to put throughout the text the times that each experiment has been done. In my opinion, many of the results presented show slight differences between the different treatments and only the fact that the result is repeated in a sufficient number of experiments gives it credibility.

Response 2: Thanks again for the helpful comments. Accordingly, we have revised the manuscript by adding in the repetition times of the experiments in the Materials and Methods section (4.3, 4.4, 4.5, and 4.7).

Reviewer 2 Report

In this article, authors developed a novel bi-specific aptamer (BBiApt) for cancer immunotherapy applications. The multivalent BBiApt consisted of two MUC1 aptamers and two CD16 aptamers, exhibiting advantages over either monovalent MUC1 or CD16 aptamer in inhibiting cell proliferation of MUC1+ tumor cells by recruiting CD16+ immunocytes to tumor sites.  

This work is interesting. Although it’s not the first study to develop bi-specific aptamers for cancer immunotherapy1, this work constructed the first bivalent bi-specific aptamer with two MUC1 and two CD16 aptamers for dual targeting in MUC1+ tumors. An innovative phagemid method was used to construct this large aptamer hybrid. Moreover, this study is logically designed. To evaluate the performance of this bi-specific aptamer, a good amount of scientific evidence have been provided by authors from characteristic and functional experiments. As a whole, this study broadens the application of nucleic acid aptamers in cancer immunotherapy. Meanwhile, the article basically meets the submission criteria of Molecules and its research topic can fit in with the scope of Journal Molecules. Therefore, I recommend publication in Molecules after some minor revisions:

1.      Introduction section,

a)        Line 59-60, authors could add more detail of CD19 for a wider audience – for which cancers can CD19 be a biomarker?

b)        Line 83-84, it appears quite sudden to mention multivalent aptamers here. This sentence could start by an introduction of current monovalent aptamers as a leadin.  

c)        Line 85-86, why choose CD16 as markers of immunocytes? Why not CD3 that’s targeted by Blinatumomab (a BiAb mentioned in line 53 for cancer immunotherapy)?

2.      Results section,

a)        Line 103-105, why the length of spacer is 60 bases? Have you ever tried other lengths? Why one spacer links two different aptamers? Can aptamers linked by one spacer be the same type?

b)        Line 124, in the figure 2, should size unit of the DNA ladder be bp?. “nt” is used for the single-stranded DNA.

c)        Line 148-149, in figure 3, authors only mentioned the way they used to hybridize FAM-probe to the bi-specific aptamer, but what’s the approach used for monovalent aptamers? Will the hybridization method affect the binding affinity of aptamers to target cells?

d)        Line 212, in figure 7, please indicate figure 7A or figure 7B.

e)        Please clarify repetitions of all experiments to rigorously support quantification data for figure 3-5 and other figures

3.      Methods section

Please clarify the amount of aptamers added in the cellular binding assay and trypsin digestion experiment.

References

1.      Pastor, F.; Kolonias, D.; McNamara, J.O., 2nd; Gilboa, E. Targeting 4-1BB costimulation to disseminated tumor lesions with bi-specific oligonucleotide aptamers. Molecular therapy: the journal of the American Society of Gene Therapy 2011, 19, 1878-1886.

Author Response

Response to Reviewer 2 Comments

Point 1: Introduction section,

a)    Line 59-60, authors could add more detail of CD19 for a wider audience – for which cancers can CD19 be a biomarker?

b)    Line 83-84, it appears quite sudden to mention multivalent aptamers here. This sentence could start by an introduction of current monovalent aptamers as a leadin. 

c)    Line 85-86, why choose CD16 as markers of immunocytes? Why not CD3 that’s targeted by Blinatumomab (a BiAb mentioned in line 53 for cancer immunotherapy)?

Response 1: Thank you very much for the valuable comments that help us to improve the manuscript. Accordingly, we have revised the manuscript per your suggestions

a)    More background details on CD19 were added in the Introduction section, line 58-62. CD19 is mainly overexpressed in hematologic malignancies, including most B-cell lymphomas, B-cell acute lymphocytic leukemia, and B-cell chronic lymphocytic leukemia.

b)    We have revised Introduction section (line 84-86) accordingly. Basically, monovalent aptamers sometimes do not have satisfactory targeting-binding capability, and multivalent aptamers may be used to improve the binding avidity.

c)    Thanks for the excellent suggestion. Although previous studies have reported both CD3 aptamer [1] and CD16 aptamer [2], the CD3 aptamer does not function well because it was selected using a competition procedure. Specifically, CD3+ cells were incubated with DNA library first. Then CD3 monoclonal antibodies were added to the cell mixture to compete with aptamers. The DNA sequences outcompeted by antibody were next PCR-amplified. Because of this particular aptamer selection process, the target-affinity of the CD3 aptamer cannot compete with that of the CD3 monoclonal antibody. This means the CD3 aptamer probably has relatively poor targeting-binding ability. We tested the CD3 aptamer with flow cytometry and found that it indeed had very poor affinity for CD3-positive cells. As a result, this aptamer was abandoned for this project. Nevertheless, in the future, if another CD3aptamer with good function is developed, it is entirely reasonable to target CD3 with bispecific aptamers. 

The CD16 aptamer was selected through the traditional SELEX procedure, and has reliable NK-binding capability. Researchers have minimized the aptamer to a 34nt sequence, which is convenient for applications. Moreover, the Kd of this 34nt aptamer was 47nM, which was very impressive for such a short aptamer. This aptamer also showed good target-binding features when being evaluated by flow cytometry in our research.

Point 2: Results section,

a)    Line 103-105, why the length of spacer is 60 bases? Have you ever tried other lengths? Why one spacer links two different aptamers? Can aptamers linked by one spacer be the same type?

b)    Line 124, in the figure 2, should size unit of the DNA ladder be bp?. “nt” is used for the single-stranded DNA.

c)    Line 148-149, in figure 3, authors only mentioned the way they used to hybridize FAM-probe to the bi-specific aptamer, but what’s the approach used for monovalent aptamers? Will the hybridization method affect the binding affinity of aptamers to target cells?

d)    Line 212, in figure 7, please indicate figure 7A or figure 7B.

e)    Please clarify repetitions of all experiments to rigorously support quantification data for figure 3-5 and other figures

Response 2: Thank you very much for the excellent comments.

a)    A 60-base DNA spacer was adopted because we want to provide enough space between the MUC1 aptamer and the CD16 aptamer, so that they do not interfere with each other’s functions. We speculate that a spacer longer than 60-base will also work well. However, it would be difficult to chemically synthesize the DNA sequence if the spacer was too long. As a result, other lengths of the spacer were not tried in this research.

By linking two different aptamers with one spacer, any two adjacent aptamers may serve as a functional unit for bispecific binding. Hence we adopted this approach. We did not construct an even longer DNA structure, because the entire length of the DNA is restricted to 800-1000 bases due to the Sanger sequencing limitation.

b)    Thanks for your keen comment. We adopt the method developed in the Cosimo Ducani’s study [3]. DNA ladder was denatured in the polyacrylamide gel with 8M urea, so the size unit was nt for single-strand DNA.

c)    The length of monovalent aptamers is relatively short, so monovalent aptamers can be FAM-labeled during chemical synthesis. Because there is a spacer between the FAM and the DNA, the aptamer’s binding affinity is usually not affected by the fluorescent labeling.

d)    We appreciate your sharp observation. Figure 7A and figure 7B were indicated in the revised manuscript.

e)    Experiments including flow cytometry and cytotoxic assays were repeated thrice during this research. Accordingly, we added the repetition clarification in the Materials and Methods section (4.3, 4.4, 4.5, and 4.7).

Point 3: Methods section

Please clarify the amount of aptamers added in the cellular binding assay and trypsin digestion experiment.

Response 3: Thanks for the valuable comments. Accordingly, we have revised the method section (4.3 and 4.5) of the manuscript. The amount of aptamers amount of aptamers is 40pmol, which is the same as that used in competition assays.

References

1.         Zumrut, H.E.; Ara, M.N.; Maio, G.E.; Van, N.A.; Batool, S.; Mallikaratchy, P.R. Ligand-guided selection of aptamers against T-cell Receptor-cluster of differentiation 3 (TCR-CD3) expressed on Jurkat.E6 cells. Analytical biochemistry 2016, 512, 1-7, doi:10.1016/j.ab.2016.08.007.

2.         Boltz, A.; Piater, B.; Toleikis, L.; Guenther, R.; Kolmar, H.; Hock, B. Bi-specific aptamers mediating tumor cell lysis. The Journal of biological chemistry 2011, 286, 21896-21905, doi:10.1074/jbc.M111.238261.

3.         Ducani, C.; Kaul, C.; Moche, M.; Shih, W.M.; Hogberg, B. Enzymatic production of 'monoclonal stoichiometric' single-stranded DNA oligonucleotides. Nature methods 2013, 10, 647-652, doi:10.1038/nmeth.2503.
